# Amyloid β, Lipid Metabolism, Basal Cholinergic System, and Therapeutics in Alzheimer’s Disease

**DOI:** 10.3390/ijms232012092

**Published:** 2022-10-11

**Authors:** Victoria Campos-Peña, Pavel Pichardo-Rojas, Talía Sánchez-Barbosa, Emma Ortíz-Islas, Citlali Ekaterina Rodríguez-Pérez, Pedro Montes, Gerardo Ramos-Palacios, Daniela Silva-Adaya, Rafael Valencia-Quintana, Jorge Francisco Cerna-Cortes, Danira Toral-Rios

**Affiliations:** 1Experimental Laboratory of Neurodegenerative Diseases, National Institute of Neurology and Neurosurgery Manuel Velasco Suárez, Mexico City 14269, Mexico; 2Facultad de Ciencias de la Salud, Autonomous University of Baja California (UABC), Valle de las Palmas, Mexicali 21100, Mexico; 3Department of Molecular Biomedicine, Center for Research and Advanced Studies of the National Polytechnic Institute, Mexico City 07360, Mexico; 4Laboratory of Molecular Neuropharmacology and Nanotechnology, National Institute of Neurology and Neurosurgery Manuel Velasco Suárez, Mexico City 14269, Mexico; 5Neuroimmunoendocrinology Laboratory, National Institute of Neurology and Neurosurgery Manuel Velasco Suárez, Mexico City 14269, Mexico; 6Department of Neurology and Neurosurgery, Montreal Neurological Institute, McGill University, Montréal, QC H3A 0G4, Canada; 7“Rafael Villalobos-Pietrini” Laboratory of Genomic Toxicology and Environmental Chemistry, Faculty of Agro-biology, Autonomous University of Tlaxcala, Apizaco 90316, Mexico; 8Department of Microbiology, National School of Biological Sciences, National Polytechnic Institute, Mexico City 11350, Mexico; 9Department of Psychiatry, Washington University School of Medicine, St. Louis, MO 63110, USA

**Keywords:** amyloid β, cholesterol, lipid rafts, cholinergic system, Alzheimer’s disease

## Abstract

The presence of insoluble aggregates of amyloid β (Aβ) in the form of neuritic plaques (NPs) is one of the main features that define Alzheimer’s disease. Studies have suggested that the accumulation of these peptides in the brain significantly contributes to extensive neuronal loss. Furthermore, the content and distribution of cholesterol in the membrane have been shown to have an important effect on the production and subsequent accumulation of Aβ peptides in the plasma membrane, contributing to dysfunction and neuronal death. The monomeric forms of these membrane-bound peptides undergo several conformational changes, ranging from oligomeric forms to beta-sheet structures, each presenting different levels of toxicity. Aβ peptides can be internalized by particular receptors and trigger changes from Tau phosphorylation to alterations in cognitive function, through dysfunction of the cholinergic system. The goal of this review is to summarize the current knowledge on the role of lipids in Alzheimer’s disease and their relationship with the basal cholinergic system, as well as potential disease-modifying therapies.

## 1. Introduction

Alzheimer’s disease (AD) is the most common type of dementia in aging and characterized by progressive loss of memory and other cognitive functions. The extracellular accumulation of neuritic plaques (NPs) and intracellular accumulation of neurofibrillary tangles (NFTs) are the histopathological markers that define AD and significantly contribute to synaptic defects and neurodegeneration [1,2,3]. NFTs are composed of tau, a microtubule-binding protein responsible for giving stability to the neuronal cytoskeleton. In pathological conditions, tau loses its affinity to microtubules and aggregates, forming paired helical filaments (PHF), which are the fundamental units of NFTs [4,5,6,7]. 

Although the presence of NFT remains strictly related to the degree of dementia in AD patients, studies have proposed that NPs play a fundamental role in the development of pathology. NPs are constituted of amyloid-β (Aβ) which is the result of the processing of the Amyloid-β Precursor Protein (AβPP) by β-and γ-secretases [8,9,10,11,12]. Like AβPP, both β-and γ-secretases are integral membrane proteins, and their function depends mainly on the levels of lipids present in the membrane. In fact, it has been reported that cholesterol homeostasis can influence amyloid production [13,14,15]. In addition to cholesterol, other lipids embedded in the membrane, such as phosphoglycerides, sphingolipids, cardiolipin, and phosphatidylinositol 4,5-bisphosphate (PIP2), modify the function of multiple transmembrane proteins, controlling many of the metabolic functions of the cell [13,16]. Such is the case of the nicotinic acetylcholine receptor (nAChR). Studies have indicated that cholesterol determines the function and stability of this receptor in the plasma membrane, and the reduction of cell-surface cholesterol produces an alteration of the organization of this receptor [13,17,18].

## 2. Amyloid β Peptide

Aβ is a peptide of 36 to 43 amino acid residues in length, originating from proteolytic cleavage of a type I transmembrane glycoprotein, known as AβPP. The cut is carried out by the sequential action of two proteases, first cleaved by β-secretase (a transmembrane aspartic metalloproteinase, known as BACE1) at the N-terminus and second cleaved by γ-secretase at the C-terminus; resulting in a large ectodomain termed APPβ, an intracellular domain (AICD) and the secretion of Aβ species of different sizes (Aβ36, Aβ38, Aβ39, Aβ40, Aβ42, Aβ43) [19,20,21,22,23,24]. The most common Aβ found aggregated in neuritic plaques of the brain of Alzheimer’s patients is the Aβ42 [25,26], which also shows a high propensity for spontaneous accumulation in solution at a micromolar concentration [27,28]. In addition to full-length Aβ1–40 and Aβ1–42 peptides, several N-terminally truncated Aβ forms have been described [29,30,31]. These isoforms (Aβ_2–42_, Aβ_7–42_, Aβ_8–42_, or Aβ_9–42_) play an important role in neurodegeneration due to their high abundance, amyloidogenic propensity, and toxicity. In vivo studies have indicated that intraventricular injections of Aβ(4–42) in WT mice tend to affect working memory [32]. Aβ aggregation occurs on the surfaces of neurons and cerebral blood vessels, forming amyloid fibrils and deposits [28,33,34]. However, the progression and severity of the disease is not associated with the number of NPs [35,36,37]. Recently, small soluble Aβ-oligomer intermediates have been found elevated in early AD patients and implicated in neurotoxicity, synaptic dysfunction, and memory loss. These features have been observed in different brain locations highlighting the nucleus basalis of Meynert in the basal forebrain and hippocampus [38,39].

In healthy brains, around 80% of Aβ peptides are Aβ40; in contrast, plaques are not evidently associated with AD. Aβ42 is produced in low picomolar concentrations. These non-toxic concentrations have physiological implications in synaptic plasticity and memory, among others [40]. 

## 3. Aβ/APP-Cellular Membrane and Lipid Influence

Cellular membranes do not only act as physical barriers, but also are dynamic structures where their components (lipids, carbohydrates, and proteins) determine the physicochemical properties and functions. The membranes composition is different between cell types, throughout life, and degenerative diseases; and can also suffer changes in response to the environment, altering cellular communication and protein-lipid interactions [41,42].

The lipid composition of cellular membranes is diverse and mainly constituted by phospholipids that are organized in a bilayer of an amphipathic nature; with phosphatidylethanolamine, phosphatidylcholine, phosphatidylserine, and sphingomyelin comprising more than half the total of lipids in most mammalian membranes. The second more abundant class of lipids is glycolipids and sterols [43,44,45,46,47]. 

Membrane fluidity is given by the acyl chain composition of membrane lipids, while straight saturated acyl chains are efficiently packed closely together. The fold in the hydrocarbon chain of unsaturated acyl chains prevents efficient packing and allows membrane fluidity [48,49]. 

The AβPP processing by secretases that occurs in the membrane is strongly associated with their physical-chemical state. It has been reported that an increase in the membrane fluidity favors the cleavage by α-secretase (preventing the Aβ peptides formation), while a reduction of the fluidity promotes the activity of γ-secretase, inducing the amyloidogenic processing of AβPP and increasing the overall production of longer forms of Aβ [23,50,51,52,53]. 

On the other hand, the interaction between Aβ and phospholipids induces a structural transition from random coil to β-sheet in Aβ40 and Aβ42 peptides, increases the local peptide aggregation, and accelerates fibril formation. Moreover, Aβ aggregates development occurring inside the membranes and causes the exposition of hydrophobic Aβ sites, which decrease membrane fluidity [54,55,56]. 

Cholesterol also contributes to membrane fluidity, promoting the packing of lipids, and is suggested to have a high affinity for Aβ, particularly by the fraction Aβ25–35 that could penetrate in the cholesterol monolayer [44,45,57]. Also, cholesterol participates in the amyloid pore’s formation, induced by different Aβ peptides (Aβ1–42, Aβ22–35, and Aβ25–35) [58,59,60]. In addition, multiple evidence has indicated that the amyloidogenic processing of AβPP occurs in specialized domains of the membrane known as lipid rafts [61,62]. The presence of high concentrations of AβPP and γ-secretase in the lipid rafts domains suggests their role in biogenesis and the accumulation of Aβ [63,64,65,66].

## 4. Lipid Rafts

Lipid rafts are defined as small heterogeneous domains (10–200 nm), highly dynamic, and enriched with sphingolipids and sterols that participate in the compartmentalization of cellular processes [67,68,69]. The main sphingolipids present in lipid rafts are ganglioside 1 and 2 (GM1, GM2 respectively). They serve as a platform for protein-lipid and protein-protein interactions and cellular signaling events, in addition to participating in protein and lipid trafficking in the secretory and endocytic pathways by regulating vesicle sorting and formation [67,68,70,71]. It is important to note that these lipid rafts are not static since they move between the lipids of the membrane and can be associated with other rafts to form larger raft domains [72,73]. 

Two types of lipid rafts have been described: the planar lipid raft and invaginated lipid raft, called caveolae, which is associated with the protein caveolin-1 (CAV1). Caveolae are involved in the endocytosis of different proteins and participate in signal transduction [74]. In neurons, lipid rafts exist exclusively in the planar form and preferentially accumulate on somatic and axonal membranes, and flotillin proteins are an indispensable requirement for their formation [75]. They belong to the prohibitin family (PHB), located in the inner surface of the plasma membrane, and are hydrophobic myristoylated and palmitoylated proteins. In the human brain, they are abundantly expressed in pyramidal neurons in the cortex, as well as in the astrocytes of the white matter. Researchers have observed that its expression increases in subjects with Down syndrome (which are prone to develop earlier onset dementia) and AD patients [74,76]. Furthermore, flotillins participate in axonal regeneration [77], membrane and vesicular trafficking [78], cell migration, signal transduction, and endocytosis, which depends on the levels of phosphorylation and S-palmitoylation of the protein [78,79,80]. 

Dysregulation in membrane components, specifically in lipid composition and homeostasis, plays an essential role in the development of neurodegenerative disorders such as AD. In this sense, lipid rafts are key factors for signaling processes, and their alteration could affect the signaling mechanisms, since it can generate an abnormal distribution and aggregation of proteins that favor the neurodegenerative process.

Several studies have pointed out that β, γ-secretase, AβPP, as well as Aβ oligomers, localize to lipid rafts, causing toxicity and interfering with signal transduction [81,82,83,84,85]. In fact, lipid rafts play an important role in the proteolytic processing of AβPP. It has also been shown that high levels of phosphorylated tau and ApoE were observed in the lipid rafts of aged Tg2576 mice and AD patients [84]. This finding is relevant suggesting that the interaction of Aβ and ApoE in the lipid rafts plays an important role in the formation of neurofibrillary tangles [84]. In addition, lipid alterations observed in neuronal membranes of AD patients have also been related to the detriment of ganglioside metabolism. Different possible links between AD and gangliosides have been proposed; for example, it is known that Aβ binds and inhibits the activity of GD3-synthase, the principal enzyme that converts α-series gangliosides into brain-specific β-series gangliosides. Furthermore, KO-ganglioside mice showed similar symptomatology to AD [70].

## 5. Cholesterol and Pathogenesis of AD

Although many authors have suggested that cholesterol plays a very important role in the development of AD, its participation in neurodegenerative processes remains controversial. Cholesterol is the main lipid required to maintain the homeostasis of cell membranes [86]. Its content in the lipid bilayer allows the permeability and fluidity of the membranes; and importantly, participates in cell signaling processes such as synapse formation [87], intracellular trafficking and sorting of proteins [88], biogenesis of synaptic vesicles [89], synaptic plasticity, and neuronal degeneration [90]. Cholesterol also plays a role as a precursor to all neurosteroids (steroid hormones synthesized locally in the nervous system), as well as the biosynthesis of oxysterols, which participate in many metabolic pathways [91,92]. Cellular membranes are asymmetric and cholesterol is non-randomly distributed in specific domains. The cytosolic side of plasma membranes is mainly composed of abundant amounts of phosphatidylethanolamines, a lower proportion of phosphatidylcholines (PC), and charged phosphatidylserines. The extracellular side, on the other hand, is composed of phosphatidylcholine and sphingomyelin [93,94,95]. Cholesterol is present on both sides of the membrane; and within the membrane, it interacts with sphingolipids as well as phospholipids to form complexes [96,97]. Also, cholesterol is the main constituent of lipid rafts [96]. 

The brain is the most cholesterol-rich organ and can sequester up to 25% of the total cholesterol of the whole body [98,99]. Unlike the rest of the body, in the brain, cholesterol is not esterified, and it is also possible to find small amounts of desmosterol and cholesteryl-ester [99]. It is an essential component of cellular membranes and myelin; approximately between 70 and 80% of the cholesterol in the adult brain is found in myelin sheaths; the rest is found in the plasma membranes of astrocytes and neurons to preserve their morphology and synaptic transmission [99,100]. The cholesterol present in the brain derives from de novo synthesis in the central nervous system, since circulating cholesterol is unable to cross the blood-brain barrier (BBB) [101]. Cholesterol synthesis occurs mainly in the endoplasmic reticulum (ER) of astrocytes and neurons. The recently synthesized cholesterol is transported from the ER to the plasma membrane by Golgi-independent and Golgi-dependent mechanisms [102,103]. Cholesterol redistribution between the different subcellular compartments is carried out through vesicle-mediated inter-organelle transport and protein-mediated monomeric transfer within the aqueous cytoplasm [100]. In neurons, cholesterol is synthesized in soma and transported to axons [104].

Morphology and healthy neuronal function depend on the availability of cholesterol. The cholesterol content in the neurons is correctly regulated by a feedback mechanism that balances biosynthesis, import, and excretion [100]. It is important to note that both cholesterol deficiency and excess can significantly damage the function of neuronal cells, leading to neurodegeneration [45,105]. Cholesterol levels can be measured by cells through membrane-bound transcription factors known as sterol regulatory element-binding proteins (SREBP); which regulate the transcription of genes that encode enzymes for the biosynthesis of cholesterol and fatty acids, as well as lipoprotein receptors [106,107]. SREBPs exit the ER in a complex with SREBP cleavage-activating protein (SCAP), to move to the Golgi, where SREBP is cleaved, releasing a fragment that activates genes for lipid biosynthesis. The “excess” of sterols blocks their exit from the ER, preventing cleavage, decreasing transcription, and controlling lipid synthesis [106,107]. 

It is important to mention that cholesterol metabolism is also strictly regulated by development; their synthesis is particularly high in the developing brain, but it declines during adulthood [99,108]. High cholesterol synthesis in young subjects could be the result of elevated hippocampal activity. During aging, oligodendrocytes lose their ability to produce the cholesterol required for myelination, leading to a decrease in neuronal plasticity. However, a constant level of total cholesterol was observed in the elderly, suggesting that under normal conditions apolipoprotein trafficking processes are not impaired [108,109]. 

Synaptogenesis requires large amounts of cholesterol and thus depends on cholesterol production by glial cells [110]. The availability of cholesterol is also known to limit the development of synapses. In the same way, low expressions of cholesteryl esters and free cholesterol correlates with an increase in amyloid production [111]. Cholesterol deficiency also increases the vulnerability of hippocampal glia in primary culture to glutamate-induced excitotoxicity [112], affects neurosteroids synthesis [113], and could alter cell membrane fluidity modifying their physicochemical properties. Thus, a decrease in cholesterol levels would favor neurodegenerative processes through synaptic loss, tau and amyloid pathology, and neuronal death. 

Interestingly, high levels of cholesterol in the membranes can also provoke significant alterations in neuronal homeostasis. Thus, high levels of this lipid promote amyloidogenesis, increasing AβPP processing, amyloid production since it also participates in the processes of fibrillation, transport, degradation, and clearance of Aβ peptide (Figure 1) [114,115].

One study recently reported that suppression of cholesterol synthesis in astrocytes significantly reduces amyloid and tau burden in neurons. Astrocyte-mediated cholesterol lowering in neurons induced relocalization of AβPP to membrane sites where it interacts with α-secretase, favoring the non-amyloidogenic pathway and inhibiting Aβ accumulation in membranes [116]. In addition, intracerebroventricular injection of Aβ in LDLr-/- knockout mice (a model of hypercholesterolemia) resulted in astrogliosis, oxidative imbalance, altered cell permeability within the hippocampus, memory deficits, and BBB alterations [117]. In the same way, a diet rich in cholesterol in rats generated an increase in lipid peroxidation, as well as a decrease in the production of nitric oxide. Similarly, there is a reduction in the antioxidant activity of glutathione-s-transferase (GST) and glutathione peroxidase (GSH-Px) in the brain, as well as an increase in the activity of acetylcholine-esterase (AChE) [118]. Alterations in AChE activity associated with high levels of cholesterol have been observed by other authors [117,119,120]. Recycling of cholesterol in the brain is highly ApoE-dependent since only a small amount of brain cholesterol passes through the BBB using an ApoE-dependent transport mechanism.

## 6. ApoE Lipidation

ApoE ε4 plays an important role in the pathogenesis of late-onset AD, as the most prevalent genetic risk factor. ApoE is a glycoprotein mainly synthesized in the liver and brain. In the CNS, ApoE plays a major role in cholesterol and lipid redistribution to neurons and other brain cells, but also in metabolite exchange between neurons and glial cells, synaptogenesis, maintenance of neuronal plasticity, remodeling of membranes, immune modulation, and clearance of Aβ [121,122,123,124,125].

ApoE is a plasma key lipoprotein responsible for regulating lipid metabolism and redirecting cholesterol transport, as well as their distribution through ApoE receptors and proteins associated with lipid transfer [126,127]. As in peripheral tissues, ApoE must be secreted and lipidated to accomplish its functions. Receptor binding, aggregation capacity, and affinity to Aβ are influenced by the lipidation status of ApoE.

In the CNS, ApoE can be lipidated with cholesterol, phospholipids, and triglycerides. The size, content, and degree of lipidation are dependent on its isoform. ApoE4 is poorly lipidated compared to ApoE2 and ApoE3 [122,123]. The evidence suggests that ApoE4 lipoprotein could have less cholesterol to deliver, compromising cellular functions and lipid homeostasis. Studies have suggested that increasing the lipidation of ApoE4 may be a therapeutic target for AD and other neurological disorders [121,122,128].

The ATP Binding Cassette Transporter A1 (ABCA1) is required in the ApoE lipidation process. This surface receptor promotes the efflux of cellular cholesterol to acceptor proteins such as ApoE [129]. ABCA1-/- mice show decreased cholesterol efflux and a reduction of 72% of ApoE levels in the hippocampus, 79% in the striatum, 80% in the cortex, and 98% in CSF [130,131]. Additionally, the size and structure of the ApoE-containing particles in ABCA1-/- cultivated astrocytes are different from the wild type; they are smaller (7–8 nm in diameter) and poorly lipidated (contain less ApoE and cholesterol) [130]. On the other hand, overexpression of ABCA1 increases ApoE lipidation and reduces amyloid plaque burden [122,123,129]. Although the process by which ABCA1 translocates cholesterol and lipids is still controversial, it is well accepted that binding of ABCA1 to the free form of apolipoproteins is a fundamental step. First, ABCA1 monomers diffuse freely and translocate lipids to the plasma membrane, then a conformational change produces the ABCA1 dimerization; as a dimer, ABCA1 interacts with actin filaments in the plasma membrane and gets immobilized until one apolipoprotein accepts the translocated lipids. Finally, ABCA1 dimers dissociate and resume their function as cholesterol transporters [129]. ABCG1 is another member of the ABC transporter family that has been implicated in lipid homeostasis in the brain. A functional model in the lipidation of ApoE has been proposed, in which ABCG1 culminates in the process of cholesterol efflux initiated by ABCA1 [132]. In the context of AD, the link between ABCG1 and processing of APP has remained controversial. It has been reported that CSF of ABCG1-/- mice have increased levels of Aβ compared to wild-type mice, but conversely overexpressing ABCG1 does not alter the plaque load, ApoE levels, cholesterol efflux, or cognitive performance in mouse models of AD [133]. 

Even though the mechanisms involving cholesterol redistribution to neurons are not fully understood, it is accepted that, similar to the process in hepatocytes, ApoE-lipid complexes interact with surface cell receptors. Binding to the heparan sulfate proteoglycan (HSPG) receptor is an initial step in the localization of the ApoE-lipid complex to the cell surface. Subsequent internalization into neurons includes processes mediated by the low-density lipoprotein receptor (LDLR) and low-density lipoprotein related protein (LDLRP), or through interaction with HSPG alone [122,128,134,135]. Heparan receptors blockage with heparase or lactoferrin diminishes the cholesterol efflux in cultured astrocytes; likewise, blocking LDLRP with a recombinant antibody eliminates the effects of ApoE3 on neurite outgrowth [128,136]. To eliminate intracellular cholesterol, it must be converted to oxysterols, a mechanism mediated in the CNS by cholesterol 24S-hydroxylase. 24S-hydroxycholesterol can cross the BBB, enter the peripheral circulation, and be eliminated as bile from the body [121,122].

The lipidation process also involves a series of conformational changes in ApoE. In its native structure, it is composed of two domains separated by a hinge region: the N-terminal domain (residues 1–167) is formed by a four α-helix bundle that contains the region for binding to members of the LDLR family, and the C-terminal domain (residues 206–299) composed by α-helices forming a hydrophobic core that binds to the surface of lipoproteins [123,137]. The structure adopted by ApoE once lipidated remains controversial. There are reports [138] indicating that the receptor binding region and the C-terminal domain align to interact with acyl chains of the phospholipids; while information obtained through X-ray reveals that hydrophobic faces of ApoE interact with one another and polar faces contact with phospholipids [139]. Regardless of its conformation, it is generally accepted that following lipid binding, ApoE undergoes a profound conformational change in the N-terminal domain, in which the helix bundle opens and converts hydrophobic helix-helix interactions into helix-lipid interactions [137]. This open state is needed for receptor recognition, given that it allows the formation of a greater positive electrostatic potential in the residues of the receptor-binding region and enhances binding to acidic residues in the LDLR [140].

Arg172 in ApoE plays a fundamental role in receptor recognition, given that a punctual mutation at this position leads to a drop in LDLR-binding affinity. Because Arg172 is outside the N-terminal domain and lipid-free ApoE is situated in an unstructured region, it is accepted that ApoE must first associate with lipids to get receptor recognition capabilities [121,122,123,141].

In vitro, biophysical studies have demonstrated that all isoforms of ApoE tend to form toxic aggregates when they are not lipidated (following the pattern ApoE4 > ApoE3 > ApoE2). ApoE4 is more susceptible to proteolysis; its aggregates are more toxic to cultured neurons and form at higher rates than ApoE3 and ApoE2. The fragments generated have a protofibrilar-like morphology with a high α helical content, a molecular weight of 12 to 29KDa, and the ability to evade the secretory pathway, promoting tau-phosphorylation and causing mitochondrial dysfunction [121,123,142]. It has been speculated that ApoE4 digestion may contribute to amyloid plaque formation since fragments have been found in the brains of individuals with AD [122,123].

## 7. ApoE Lipidation and Aβ

### ApoE Lipidation as a Therapeutic Target for AD

One study reported that ApoE4 binds to Aβ with a higher affinity than the other isoforms. In an AD mouse model overexpressing human ApoE4 isoform, Aβ load is increased in the brain interstitial fluid exceeding the levels observed in ApoE2-and ApoE3-expressing mice. The increased deposition of Aβ negatively correlates with its rate of clearance [143]. Thus, the efficient clearance is vital for preventing Aβ accumulation and subsequent aggregation [144]. ApoE containing lipoprotein sequesters Aβ, modulating the cellular uptake of ApoE-Aβ by receptor-mediated endocytosis. In vitro studies have demonstrated that human ApoE facilitates the internalization and binding of Aβ into several types of neuronal cells and modulates the removal of this from the brain to the systemic circulation through the BBB; both mechanisms are dependent on the ApoE isoforms [145,146,147,148,149,150,151]. 

An in vivo study showed that ApoE2-Aβ and ApoE3-Aβ complexes are cleared by the VLDL receptor (VLDLR) and LRP1 on the BBB, while ApoE4-Aβ clearing is carried out solely by VLDLR. The internalization of this complex (ApoE4-Aβ) is slower; therefore, its clearance is also slower when compared with the complexes formed by the other ApoE isoforms [150,152]. The presence of ApoE4 in microglial-like; iPSC-derived cells impaired the clearance and degradation of Aβ, compared with cells expressing ApoE3 or ApoE2 [149,153]. Cellular uptake and subsequent degradation of Aβ by astrocytes and microglial cells is a crucial Aβ clearance pathway in the brain. Different studies have indicated that receptor-mediated internalization of Aβ is affected by the presence of the APOE4 isoform in astrocytes [154,155]. 

ApoE lipidation also affects its affinity for Aβ and influences the formation of amyloid plaques in AD. The efficiency of complex formation between lipidated ApoE and Aβ follows the order ApoE2 > ApoE3 >> ApoE4 [128]. One study demonstrated in transfected RAW-264 and HEK-293 cells that native ApoE3 has a twofold higher affinity for Aβ than ApoE4, but the removal of lipid moieties results in the loss of this specificity [156]. Their results also show that lipidated ApoE molecules have a fivefold higher Aβ binding affinity than delipidated isoforms. This evidence suggests that lipidated ApoE3 enhances the clearance of amyloid β and prevents its accumulation [156]. Other in vitro analyses have pointed out that delipidated ApoE4 more rapidly promotes fibril formation [157]. Since Aβ interacts with ApoE via its C-terminal domain, it abrogates ApoE lipidation and disrupts lipid homeostasis, which may contribute to AD progression (Figure 2) [158]. 

Increasing lipidation of ApoE has been suggested as a biological target to ameliorate clinical symptoms associated with AD. Some approaches to achieve this are increasing ABCA1 activity, using Living X receptor (LXR) agonists, correcting ApoE4 structure to avoid domain interaction, overexpressing a copy of ApoE2 through an adeno-associated virus (AAV)-mediated gene therapy, or reducing ApoE4 toxic effects with immunotherapy. 

MiR-33 is a microRNA expressed throughout the brain (mainly in neurons) that modulates Aβ levels in neuronal and glial cells through ABCA1 regulation. Brain-specific miR33 inhibition increases ABCA1 levels, ApoE lipidation, and extracellular Aβ degradation while reducing Aβ secretion in the cortex of APP/PS1 mice [159]. Another approach to increase ABCA1 activity is through CS-6253, a peptide derived from the carboxyl-terminal of ApoE that is capable of oligomerizing ABCA1, increasing its levels, reversing cognitive impairment associated with ApoE4, and reducing tau hyperphosphorylation and Aβ accumulation in hippocampal neurons [160]. Silencing with siRNAs ADP-ribosylation factor 6 (ARF6), a protein that regulates the degradation and recycling of ABCA1, is another way to increase ABCA1 activity that could be beneficial for ApoE4 carriers. A study found that ApoE4 astrocytes promote greater expression of ARF6 than ApoE3, trapping ABCA1 in endosomes, thereby decreasing ABCA1 membrane expression and increasing lysosomal degradation [161]. Therefore, silencing ARF6 allows ABCA1 proper recycling to the plasma membrane, and increases cholesterol efflux and lipidation of ApoE [161].

Although LXR and Retinoid X receptor (RXR) agonists have proven efficacy to upregulate genes involved in ApoE-lipidation (like ABCA1 or ABCG1), improve Aβ clearance, and reverse memory deficits in animal models of AD, they produce unwanted peripheral effects on triglyceride levels and liver health. One study suggested that CNS-specific LXR activation could avoid systemic adverse effects [122]. 

As mentioned above, ApoE4 tends to aggregate more than ApoE3 and generates more neurotoxic C-terminal fragments after being recognized by neuron-specific proteases. The interaction domain, consisting of the ionic interaction between Arg61 and Glu255 exclusive of isoform ApoE4, is responsible for these features. Structure correctors increase CNS ApoE levels and lipid-binding capacities, decrease intracellular degradation, prevent neuropathology associated with ApoE4, and change the conformation to an ApoE3-like structure, promoting proper lipidation of ApoE and better transport of phospholipids and cholesterol. Preclinical studies to develop ApoE structure correctors as a therapeutic approach are in progress [121,122].

Targeting the non-lipidated form of ApoE4 with antibodies is another way to reduce toxicity. One study developed HAE-4, an antibody against human ApoE that specifically recognizes ApoE3 and ApoE4 in their non-lipidated conformation. When delivered centrally (by intracerebroventricular injection) or peripherally (by intraperitoneal administration), HAE-4 preferentially binds to ApoE aggregates, reducing Aβ deposition and accumulation in the brain of APPPS1–21/ApoE4 mice [162].

Converting ApoE4 homozygotes into ApoE2/4 heterozygotes via CNS genetic modification has been proposed as a strategy to balance the neurotoxicity associated with ApoE4 and the progression of AD. A study found that delivering the ApoE2 protective allele into the CSF of the lateral ventricle employing an AAV is safe and effective in non-human primates and mice. Out of the three analyzed delivery routes (intraparenchymal, intraventricular, and intracisternal), the last one achieved the widest extent of ApoE2 expression in the CNS with the least invasive surgical intervention [163]. This evidence is consistent with the findings of another study that reported that intracerebroventricular injection of AAV vectors expressing human ApoE2 in the thalamus of two mouse models of brain amyloidosis enhances ApoE lipidation, increases levels of ApoE-associated cholesterol, and decreases endogenous Aβ [164]. Moreover, brain-wide expression of Cas9 targeting the APPswe mutation in 5XFAD through an AAV capable of crossing the BBB was shown to efficiently reduce amyloid deposition across multiple brain structures, including the cortex and hippocampus [165]. In addition, individuals with high plasma cholesterol levels have a more elevated susceptibility to the development of AD, which is influenced by the presence of the ApoE e4 genotype. In fact, Aβ accumulation in neurons is regulated by cholesterol synthesis and ApoE transport from astrocytes [116].

Studies have also shown that a high-cholesterol diet activates astrocytes and potentiates the toxic effects caused by Aβ on nAChR subunit levels, such as learning and memory impairments [166]. Other authors also found that high doses of cholesterol may inhibit the α-secretase activity in APP metabolic processing and down-regulate the expression of the a7 nAChR [167]. In patients with AD, the expression of these receptors is significantly decreased. 

## 8. Cholinergic System

Although the main markers of AD are the presence of NPs and NFTs, the basal forebrain cholinergic hypo-function has an important role in the Alzheimer’s pathology, since the loss of basal cholinergic neurons is associated with severe neurodegeneration (Figure 3) [168]. It is widely known that neurotransmitters such as acetylcholine (ACh) work closely to maintain brain homeostasis. ACh has been proven essential to many brain functions, including chemical transmission at the neuromuscular junction, and regulates autonomic functions in the peripheral nervous system, and the cognitive processes in the central nervous system [169]. The literature has widely documented that cognitive impairment, observed in AD patients, correlates directly with the severity of the cholinergic deficit. The inhibition of acetylcholinesterase (AChE) allows a prolonged action of ACh on its receptors (AchRs), which leads to a process of “stabilization” in cognition, delaying the progression of the pathology [170].

### Basal Forebrain and Cognition 

Since the beginning of the twentieth century, studies have described how alterations of the BF’s cholinergic circuitry are responsible for the cognitive impairments associated with neurodegenerative diseases [171], including AD; and even seen in normal aging [172,173]. Currently, BF is intimately involved in learning, attention, and memory processes. The BF and, in particular, the cholinergic circuitry therein exerts strong modulatory influence at the molecular, cellular, and systems levels. Pharmacological perturbations and viral manipulations of BF circuitry in AD models combined with optogenetics tools have enabled the precise dissection of the cholinergic system in behaving mice [174]. Using a combined optogenetic and viral tracing approach in a transgenic model of AD (Tg2576-APP^swe^ mice), it was established that ChAT+ neurons (choline acethyltransferase neurons) + in the diagonal band of Broca directly innervates immature neurons of the dorsal hippocampus and that this connection is essential for cholinergic transmission and hippocampal neuron survival (Figure 4).

Importantly, this effect was blocked when the M1 cholinergic receptor was virally ablated in dorsal hippocampal neurons, highlighting the importance of cholinergic innervation for spatial memory formation. These mice also display synaptic defects in the density of ChAT labeled terminals compared to non-AD controls [175]. To date, the best pathophysiological correlate of cognitive decline is the early synaptopathy that precedes neurodegeneration and is recapitulated across AD models [176]. The development of genetic models has advanced our understanding of converging mechanisms of Aβ pathology and cholinergic degeneration. In 5XFAD mice bearing five familial AD mutations, detailed unbiased stereological analysis and 3D reconstruction of ChAT fibers and ChAT + neurons at different stages showed the selective fiber damage to cortical areas innervated by the basal forebrain cholinergic circuitry preceding degeneration of other structures [177], and coinciding with early cortical deposition of amyloid in this model [178]. Selective damage to cholinergic neurons using intracerebroventricular delivery of immunotoxin in the APPswe/PS1dE9 mice exacerbated plaque burden, in comparison to WT animals, and worsened working memory [179], a cardinal feature of early AD and Mild Alzheimer’s Disease Dementia.

An often-overlooked function of cholinergic innervation is the regulation of cerebral blood flow. Cholinergic innervation of cortical blood vessels plays a fundamental role in regulating neurovascular coupling [180], an intricate process required for maintaining proper cognitive function and which serves as the basis of functional neuroimaging studies. Chronic depletion of acetylcholine leads to altered neurovascular unit coupling in response to stimuli in awake behaving mice [181]. Imaging studies in humans have confirmed dysregulation of cerebral blood flow [182], and BBB breakdown in the hippocampus and medial temporal lobe of ApoE4 carriers, which correlates with cognitive decline [183]. These vascular deficits also correlate with early synaptic dysfunction across transgenic AD models and are increasingly recognized as key drivers of pathology [184,185,186].

In addition to the neurotransmission and neurovascular defects, an important consequence of cholinergic dysregulation is neuronal hyperexcitability. This view is strengthened by the recent demonstration of cortical innervation of acetylcholine projection neurons to inhibitory neurons [187]. The early dysfunction in this neuromodulatory loop could lead to cognitive deficits. It will be important for future research to characterize whether the initiation of pathological deposition of Aβ differentially affects neocortical inhibitory circuits. The functional significance is further highlighted by work using deep brain stimulation targeting the NBM in primates with positive results on cognition and memory [188]. The intersection of basal cholinergic lesions and the initiating factors contributing to the development of a beta pathology have been detailed in experimental models, pointing to a possible synergistic effect in genetically vulnerable populations.

## 9. nAChRs, Lipids and Aβ

The different pathogenic mechanisms of the AD may be explained based on the relationship with the Aβ cascade, which postulates that the presence of amyloid is capable of promoting tau hyperphosphorylation, which leads to the formation of neurofibrillary tangles and oxidative stress, as well as alterations in the cholinergic system [171,189,190,191]. In this sense, it has been proposed that the interaction between Aβ and the plasma membrane correlates with the degree of toxicity, and a reduction in cholesterol levels decreases the toxic effect generated by Aβ [15]. Additionally, the binding of Aβ to the membrane alters the structure and fluidity of the lipid microenvironment, generating damage to the function of membrane receptors, including the α7 nicotinic acetylcholine receptor (α7 nAChR) [192]. It has been postulated that lipid composition influences the conformational state of nicotinic acetylcholine receptors (nAChRs); thus, membrane fluidity or some other property is able to modulate the balance between resting and desensitizing states [193]. In fact, cholesterol is one of the main lipids that determines the function and stability of nAChR in the plasma membrane, as well as its supramolecular organization [194]. In this context, the interaction between Aβ and nAChR could indirectly occur; that is, the damage generated by the interaction of Aβ with membrane lipids results in an alteration in the function of nAChR that leads to cognitive impairment [192]. This effect has been demonstrated in a rat model, in which after intracerebroventricular injection of Aβ1–42, peptide deposits, and astrocytic activation, as well as a decrease in nAChR α7 and α4 subunit levels, were observed. The administration of a diet rich in cholesterol enhances the toxic effects generated by the peptide on receptor subunits and increases learning and memory impairment [166]. In this way, chronic treatment of hippocampal neurons with lovastatin and methyl-β-cyclodextrin inhibit Aβ generation [195]. 

In the CNS, neuronal nAChR is associated with lipid rafts, which play a significant role in the formation and maintenance of the α7nAChR. The extraction of cholesterol with methyl-β-cyclodextrin resulted in the dispersion of lipid rafts and the redistribution of α7nAChR in small groups on the cell surface, in addition to causing the depolymerization of F-actin filaments; demonstrating that the integrity of the components of the lipid rafts are essential to maintain the structure of the receptors [196,197]. The proposal that cholesterol depletion reduces the number of cell-surface α7AChRs has been widely documented [198,199]. Studies demonstrated that under Chol− conditions, the AchR is internalized through a pathway that depends on the activity of small GTPases (Arf6 and Rac1) and the effector phospholipase D downstream of Arf6 [194,199]. It is noteworthy that under the conditions in which cholesterol levels are normal, this internalization pathway does not include endocytosis of AchR. [199,200]. Finally, other studies showed that the inhibition of cholesterol synthesis generates desensitization kinetics without affecting the current amplitude of the α7 nAChRs; improving cholinergic function and thus cognition and memory in animal models [201,202].

## 10. Therapy

Encouraged by the need to find an effective treatment against AD, many inhibitors of Aβ aggregation and cytotoxicity such as small molecules, peptides, proteins, antibodies, and alternative therapies based on nanotechnology began to be studied and tested. At present, treatments to control cognitive impairment in AD are based on neurotransmitter or enzyme substitution, providing symptomatic benefits. These treatments include acetylcholinesterase inhibitors, antioxidants, amyloid-targeting drugs, nerve growth factors, secretase inhibitors, Aβ vaccines, and statins [203,204,205,206]. Recently, the FDA approved five medications for the treatment of AD: tacrine, donepezil (DPZ), rivastigmine, galantamine, and memantine. The first four are acetylcholinesterase inhibitors, while the last one is an N-methyl-D-aspartate receptor antagonist [207,208]. Alternative approaches for the treatment and diagnosis of AD include the use of nanotherapy that targets the main systems and molecules implicated in the pathogenesis of the disease.

### 10.1. Statins

Currently, 3-hydroxy-3-methylglutaryl coenzyme A (HMG-CoA) reductase inhibitors, better known as statins, are the most effective lipid-lowering drugs. In general, statin treatment reduces the amount of cholesterol in the brain membranes. In addition, statins have anti-inflammatory, antioxidant, and neuroprotective effects and are classified into two categories: hydrophilic (fluvastatin, mevastatin, and pravastatin) and lipophilic (simvastatin, lovastatin, atorvastatin) [209]. In vitro assays have shown that the use of these drugs decreases Aβ levels, and it has been proposed that they activate ADAM10 [210] and increase the activity of phospholipid transporters (PLTP), which results in the reduction of p-tau181 (Table 1) [211,212,213,214,215,216,217,218,219,220,221].

Although preclinical trials have indicated that the use of statins confers protection against AD development, clinical trials remain controversial [188]. Some authors have pointed out that the early use of lipophilic statins is associated with a significant decrease in the progression of AD in those with mild-moderate AD, and its use could be beneficial in patients with AD since it modifies cognitive deterioration mainly in those subjects homozygous for ApoE4 [222,223]. 

In fact, statins can also prevent the neuroinflammatory effects that favor neurodegeneration. Using an in vitro model of lipopolysaccharide (LPS), highly aggressively proliferating immortalized cells (HAPI microglia) were treated with LPS (0.1 μg/mL; 24 h), which produced a very high release of reactive oxygen species (ROS), nitric oxide, and IL-1β, TNF-α and PGE2. This conditioned medium was transferred to neuroblastoma cultures (SH-SY5Y), to determine different parameters such as cell viability, mitochondrial membrane permeability, apoptosis, autophagy and ROS production; showing that treatment with atorvastatin, pravastatin and rosuvastatin protects SH-SY5Y cells from damage generated by LPS [224] (Figure 5).

### 10.2. AChE Inhibitors

Currently, use of AChE inhibitors (AChEI) is the main pharmacological therapy used in patients with AD, which provides a remarkable improvement in the cognition of these patients. The main inhibitors used are donepezil, rivastigmine, and galantamine. The efficacy of each of them has been extensively studied in several studies, both in animal models and clinical trials. Its use has shown that it has a beneficial effect in all stages of the disease and could also have a neuroprotective effect by increasing cell viability and reducing neuronal death, as well as the presence of inflammatory mediators [225,226].

#### 10.2.1. Tacrine (TCR)

Tacrine, the first drug approved by the FDA in 1993 to treat AD, is an acridine derivative. Despite being an excellent AChE/BuChE inhibitor, its use has been discontinued due to side effects and toxicity [227]. However, analogues of tacrine have been shown to have fewer side effects. For example, a study that synthesized and described the tetracyclic tacrine analogues containing pyrano [2,3-c]pyrazole (1a–e) showed that the compound containing a 3,4-dimethoxyphenyl group (compound 1d) protected neurons from oxidative stress [228]. In the same way, another group evaluated eight new racemic 3-methyl-4-aryl-2,4,6,7,8,9-hexahydropyrazolo [4′,3′:5,6]pyrano [2,3-b]quinolin-5-amines (pyranopyrazolotacrines, PPT). Their results showed a selective inhibition of the acetylcholinesterase and nonhepatotoxic effect [229].

#### 10.2.2. Donepezil (DPZ)

Donepezil is widely prescribed and has been considered a first-line treatment for AD since 1996 with selective activity for AChE. Several studies have indicated that the use of donepezil can reverse the clinical symptoms present in AD and has also been shown to have an anti-neurodegenerative effect, promoting neuronal differentiation and neurite growth [225]. In animal models, donepezil induced cognitive and behavioral improvements [230]. AD patients treated with donepezil revealed improvements in cognitive functions as well as changes at the morphological level in structures such as the cortex and hippocampus. A study conducted a double-blind, randomized, placebo-controlled, parallel-group design study using donepezil (10 mg/day) in subjects with suspected prodromal AD. The results demonstrated a 45% reduction in hippocampal atrophy in the group of patients treated with donepezil (10 mg daily). This multicenter study is of great importance, as it shows that the use of a drug has positive effects on a morphological variable [231]. In the same way, another study showed that treatment with donepezil for one year in suspected prodromal AD patients may have an impact on cortical morphology. This study also included a Free and Cued Selective Reminding Test (FCSRT), a memory test correlated with hippocampal volume and CSF-level changes of the Alzheimer type. They found a trend showing a reduction in the thinning of the cortical areas, innervated by the medial and lateral cholinergic pathway [232]. Basal forebrain cholinergic system (BFCS) atrophy has been documented to precede the observed damage to the entorhinal cortex, as well as memory impairment. These findings show that the use of this drug helps to preserve brain morphology, mainly of structures belonging to the cholinergic circuit in prodromal AD and even in mild/moderate stages of dementia, proving its function of serving beyond just “symptomatic” treatment. It also suggests the central role of basal forebrain atrophy during the pathophysiological process in AD [233,234]. 

#### 10.2.3. Rivastigmine (RVG)

Rivastigmine was approved in 2000 for the treatment of AD patients. Unlike donepezil or galantamine (selective AChE inhibitors), rivastigmine is an inhibitor with activity for AChE and butyryl-cholinesterase (BuChE). Recent studies have reported increased levels in nerve growth factor (NGF)-induced neurite outgrowth in PC12 cells, via sigma-1 and sigma-2 receptors [235]. On the other hand, in treatment over six months with rivastigmine in a double-blind controlled study, a significant increase in hippocampal metabolism (32.5%) was observed in those subjects who responded to rivastigmine compared with the placebo (4.1%). These results demonstrate that rivastigmine not only prevents the clinical progression of symptoms but is also capable of generating a metabolic increase in structures related to memory [236,237]. Similarly, less white matter loss was found, as well as a decrease in ventricular enlargement only in women, and with no apparent effect on hippocampal volume [238,239]. It has also been reported that rivastigmine, unlike other AChEIs, is able to preserve both neocortical white matter and gray matter, and the antineurodegenerative effects of rivastigmine in the white matter could be the result of inhibition of both AChE as well as BuChE [235,239,240]. Inhibition of both cholinesterases has beneficial effects on cognition, global function, behavioral symptoms, and executive functions.

#### 10.2.4. Galantamine (GLT)

Galantamine, a selective inhibitor of AChE, is an alkaloid extracted from the *Amaryllidaceae* family. Unlike donepezil and rivastigmine, galantamine has greater selectivity in the CNS and little effect at the peripheral level; it also interacts allosterically with the nicotinic acetylcholinesterase receptor, enhancing the action of agonists at these receptors [241]. It has been proposed that the effect on these receptors contributes to the effectiveness of this drug observed in AD patients, since it increases ACh levels in the cerebral cortex by slowing the degradation of the neurotransmitter [241]. It is also known to modulate allosteric nicotinic ACh receptors, which allows a greater amount of ACh in the presynaptic terminals, while increasing glutamate and serotonin levels. Clinical trials have shown that the administration of galantamine in a single dose correlates with the effects of long-term treatment. That is, patients who show a reduction in EEG alpha and theta power, and saccadic eye movements after administration of galantamine (16 mg) are more likely to respond to treatment [242].

Although the mechanisms by which cholinesterase inhibitors can exert a neuroprotective effect have not been fully established, it has been proposed that AchEIs increase the presence of neurotrophic factors, such as NGF. The development and maintenance of the Basal forebrain (BF) cholinergic nuclei depend on NGF; therefore, the loss of function and cholinergic phenotype of BF neurons observed in AD could be due to impaired NGF-mediated trophic support [243]. Cholinergic changes occur in the early stages of the pathology, so the use of AChEI could have a beneficial and protective effect by stimulating the release and uptake of the nerve growth factor (NGF) dependent on acetylcholine, favoring anti-neurodegenerative processes in the CBF of AD patients, and stimulating upregulation of nAChRs in the CNS [239]. Other studies have also shown that these inhibitors (Donepezil, Rivastigmine, and Galantamine) promote neuronal protection by increasing cell viability, reducing neuronal death, decreasing Aβ-generated oxidative damage, and modulating inflammatory mediators (Figure 6).

However, one of the main problems in obtaining the beneficial effects of AChEI is the inadequate inhibition that occurs at the CNS level. Most of the approved drugs have an inhibition range lower than that required to obtain the beneficial effect, both clinically and anti-neurodegenerative. The main cause of this is that they are not selective inhibitors of the CNS and exert adverse effects, especially in the gastrointestinal tract.

Besides, the anatomical structure of the blood brain barrier (BBB) represents another intrinsic obstacle to effective drug delivery. The BBB is an essential defensive barrier between the CNS and the circulating blood, and comprises closely connected endothelial cells, capillary basement membranes, astrocytes, and pericytes [244]. The main functions of the BBB are to block toxic and harmful substances from the blood, maintain hemostasis of the internal brain environment, and protect the brain from noxious stimuli [245]. However, it also prevents access to most drugs, resulting in reduced bioavailability and compromising their therapeutic effects. In this way, several alternative and cutting-edge approaches have been proposed for the treatment of AD, such as nanomaterials, antibodies, stem cell transplantation, and nucleic acid insertion [246,247,248]. 

### 10.3. Nanotherapy

#### 10.3.1. Nanodelivery of AChE Inhibitors to Improve Their Effectiveness in AD Treatment

Nanotechnology is considered an area of science and engineering that employs materials, devices, and systems by controlling their size through manipulating matter at the nanometer scale with dimensions from 1 to 100 nm [249]. Nanomaterials are designed from different materials such as liposomes, lipids, albumin, polymers, carbon, ceramics, metals, and magnetic oxides [250], and their application in drug administration has been of great relevance, since they offer promising platforms for the treatment and diagnosis of several diseases [214,251,252]. Undoubtedly, the most important challenge for the treatment of central nervous system diseases is to ensure that the administered drugs can cross the BBB and carry out their therapeutic function. Some drug physicochemical properties, such as lipophilicity, ionization, molecular weight, bioavailability, metabolism, and their administration route, are essential to choosing the nanomaterial to use. Figure 7 summarizes the different nanomaterial types and in vitro, ex vivo, and in vivo research using cell lines and animal models to release anti-cholinergic drugs.

As discussed above, one of the main problems in obtaining the beneficial effect of AChEIs has been the low level found in the CNS and secondary effects observed in the gastrointestinal tract. Therefore, the use of nanomaterials has been proposed to decrease these adverse effects and enhance targeting potential with lower doses [253,254]. These nanomaterials are considered safe, capable of mimicking and modifying biological processes, and demonstrated potential as delivery carriers to treat AD [247,255]. For this purpose, experimental in vitro, ex-vivo, and in vivo procedures have been developed administering the nanomaterials by oral inhalation and intravenous routes to the animal used [254]. 

#### 10.3.2. Nanodelivery of Tacrine 

A nanostructured lipid carrier was developed as a delivery system for TCR to determine its cytotoxicity in the neuroblastoma cell line SH-SY5Y [256]. Three different nanolipid formulations loaded with TCR presented a particle size distribution below 200 nm. Release profiles indicate that after 24 h, 50.8% of TCR was released, with a subsequent decline in the time, reaching 60.7% in 72 h. Cell viability results demonstrated that the nanolipid-TCR formulations were well tolerated and safe against SH-SY5Y cells. In other work, TCR loaded in poly-(D, L)-lactide-co-glycolide (PLGA) nanoparticles with a size of approximately 70.55 nm was evaluated in vitro and in vivo [257]. The release of the different formulations followed first-order kinetics during its degradation with a TCR released in a sustained release way. The in vivo study showed that the PLGA nanoparticles in carbopol 934 NCGs have good brain efficiency after intranasal administration to rats compared with uncoated PLGA nanoparticles compared in NCGs. Histopathology ensured the safety of the intranasal administration of tested Pt-coated PLGA NPs in NCGs. Other studies on the encapsulation of TCR on poly (lactide-co-glycolide) nanoparticles evaluated pharmacodynamics studies of the nanoparticles for brain targeting and memory improvement in scopolamine-induced amnesic [258]. This work revealed that a higher transport of TCR in mice brains shows improved therapeutic efficacy of TCR in treating Alzheimer’s disease.

#### 10.3.3. Nanodelivery of Donepezil

Donepezil is one of the most widely used cholinesterase inhibitors for the treatment of AD, in mild and moderate phases. Chitosan nanoparticles, loaded with donepezil, have been used for AD treatment. Chitosan is a bioactive, biocompatible, and biodegradable polymer that can be easily designed. It was recently shown that the transformation of hyperbranched chitosan into thin films and nanofibers showed rapid and effective disintegration capacity. These nanofibers were loaded with donepezil, showing a drug release capacity of 97.03% in 30 minutes. The authors showed that by modifying the highly branched structure, the size of the particles, and the proportion of other constituents in the formulation, the drug release behavior is significantly improved, which may be favorable for the treatment of AD [259]. A study designed an administration system using solid lipid nanoparticles (SLNs), marked with rhodamine B, loaded with donepezil, and targeted with apolipoprotein E (ApoE). Their results showed that the presence of the ApoE ligand increases the uptake of these nanoparticles in human brain endothelial cell cultures and neurons. It was also possible to observe an increase in the permeability of these donepezil-loaded nanoparticles in a BBB co-culture model, suggesting that ApoE can confer or facilitate drug entry through the BBB, as well as its input to neurons, for beneficial effect on the treatment of AD [260].

Another strategy has been transdermal administration, which also aims to reduce the drawbacks or adverse effects of donepezil. For this, the development of nanostructured lipid carrier-based gels (NLC gel) capable of improving the skin delivery of donepezil free base (DPB) has been proposed. A study evaluated these DPB-NLC particles, and drug dispersion was characterized by atomic force microscopy and dynamic light scattering. Subsequently, they performed in vitro skin permeation assays and the results indicated that drug skin permeation from DPB-NLC gel was increased. They also showed that this is due not only to the potentiating effect of its components but also the lipid nanocarriers presented an additional potentiating effect to increase the flux of drugs through the skin [261].

#### 10.3.4. Nanodelivery of Rivastigmine 

Chitosan nanoparticles loaded with rivastigmine were used to target the brain through skin permeation [262]. Drug loading, ex-vivo skin permeation and kinetic studies were conducted to determine the stability over three months at different temperatures. Ex vivo studies revealed that the RVG-chitosan nanoparticles in gel formulae were not irritating to rat skin and had better skin permeation than chitosan nanoparticles’ aqueous dispersion. The chitosan nanoparticles also were capable of releasing RVG in a sustained manner and increasing the time release of the drug [262]. Another report compared liposomal formulations and observed that chitosan-coated liposomes have a better and longer release pattern than conventional liposomes. Their results also show that this optimized formulation remains stable for four weeks and there were no significant changes observed [263].

Also, RVG-chitosan nanoparticles were manufactured and administered by intranasal delivery; for this, several formulations and processing parameters were optimized to obtain chitosan nanoparticles with the desired quality. Later, the optimized formulation was coated with Eudragit to prolong the release of the drug and thereby evaluate the in vitro permeation and release study. Finally, the authors conducted a study to measure the ciliotoxicity of the coated nanoparticles and evaluate the effect on the ovine nasal mucosa, showing no toxicity to the nasal cilia of the experimental sheep nasal mucosa [264]. These developed rivastigmine nanoparticles could be used as an alternative delivery system for those drugs with low bioavailability.

#### 10.3.5. Nanodelivery of Galantamine 

The use of galantamine-complexed chitosan nanoparticles were used to prepare GH/chitosan complex nanoparticles (CX-NP2), which were administered intranasally in a rat model. The results showed that its administration for 30 days did not induce a toxic response and confirmed its localization in several brain regions. This non-invasive technique is promising since it can offer therapeutic potential for the management of AD, by recovering cholinergic transmission and restoring neuroprotection [265]. Copolymeric networks based on poly (N-isopropyl acrylamide) polymer were studied to be used as a new platform for the possible transdermal administration of GLT. The GLT was loaded into the copolymeric networks by swelling in an aqueous solution. In vitro, GLT release studies showed a considerable amount of GLT released from the different networks in the first hour. The highest release rate was found for the network obtained by redox polymerization with lower cross linker content, which was attributed to the network structure and high GLT content [266].

## 11. Supramolecular Strategies

New strategies have recently been developed, which aim not only to recognize Aβ peptides but also inhibit their fibrillation and, finally, eliminate amyloid plaques. Small molecules such as curcumin, peptides, or macrocyclic receptors have been shown to have the ability to inhibit Aβ fibrillation [267,268].

Another study designed a heteromultivalent platform that binds to Aβ 1–42 and inhibits its fibrillation. This platform is made of a co-assembly of cyclodextrin (CD) and calixarene (CA), two macrocyclic receptors with complementary recognition sites. When constructed in a 1:2 ratio, the co-assembly can disaggregate 100% of Aβ1–42 fibrils in an in vitro model. Nanoparticles of the co-assembly and polymer polyfluorene-alt-benzothiadiazole (PFBT) administered intranasally for one month to 5XFAD mice reduced plaque load and the levels of insoluble, soluble, and oligomeric Aβ1–42 in the dentate gyrus, CA1, and medial prefrontal cortex regions. Furthermore, after the treatment with the co-assembly, the expression of neprylisin was recovered and degradation of Aβ1–42 was enhanced [269].

It was also found that the 5XFAD mice treated intranasally with the CD-CA co-assembly showed reduced cognitive impairment, as well as higher dendritic spine density and increased expression of synaptophysin and postsynaptic density protein 95 in the hippocampus when compared with the 5XFAD control group. Apoptosis levels and oxidative stress were also decreased in the dentate gyrus, CA1, and medial prefrontal cortex regions in comparison with the PBS-treated 5XFAD mice.

Besides, the treatment with CD-CA-PFBT nanoparticles reduced the polarization of microglia to a neurotoxic M1 state and increased the polarization to an anti-inflammatory M2 state. Levels of iNOS, COX-2, IL-1β in hippocampus and medial prefrontal cortex regions were decreased when compared with the control counterparts [269,270].

## 12. Conclusions

Alzheimer’s disease is a progressive neurodegenerative disorder whose etiopathological mechanisms have not been fully elucidated. Its multifactorial nature makes it very difficult to manage and develop an effective therapy with adequate costs. Different types of evidence have indicated that AD could be the result of alterations in cholesterol metabolism. We have mentioned that lipids and especially cholesterol make up a very high percentage of brain weight. The lipids of the neuronal membranes play a significant role in the metabolism and homeostasis of various membrane proteins such as AβPP and AChR. In this way, the formation and accumulation of Aβ and cholinergic function are closely related to the lipid homeostasis of the membranes. Researchers have postulated that the function of the cholinergic system, as well as AβPP processing, is highly regulated by the presence of lipid rafts. Knowing how lipids interact with these proteins will allow a better understanding of the pathology and aid the development of new therapeutic strategies.

## Figures and Tables

**Figure 1 ijms-23-12092-f001:**
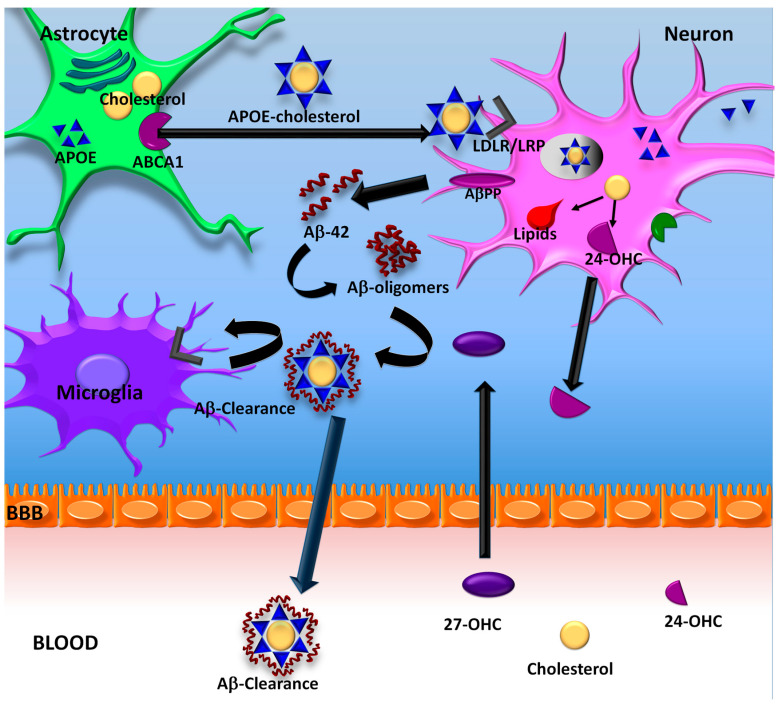
Cholesterol and Aβ clearance. Cholesterol is produced in the endoplasmic reticulum (ER) of astrocytes. Once synthesized, it binds to APOE, to be transported to the outside of the cell through ABCA1. Cholesterol-APOE complexes are internalized into neurons with the help of LDLR/LRP receptors. Altered cholesterol metabolism induces an increase in AβPP processing, leading to the release of Aβ peptides, which aggregate in oligomeric forms in the presence of lipids. Amyloid can be removed by specialized proteases within the CNS or peripherally by forming complexes with APOE-cholesterol.

**Figure 2 ijms-23-12092-f002:**
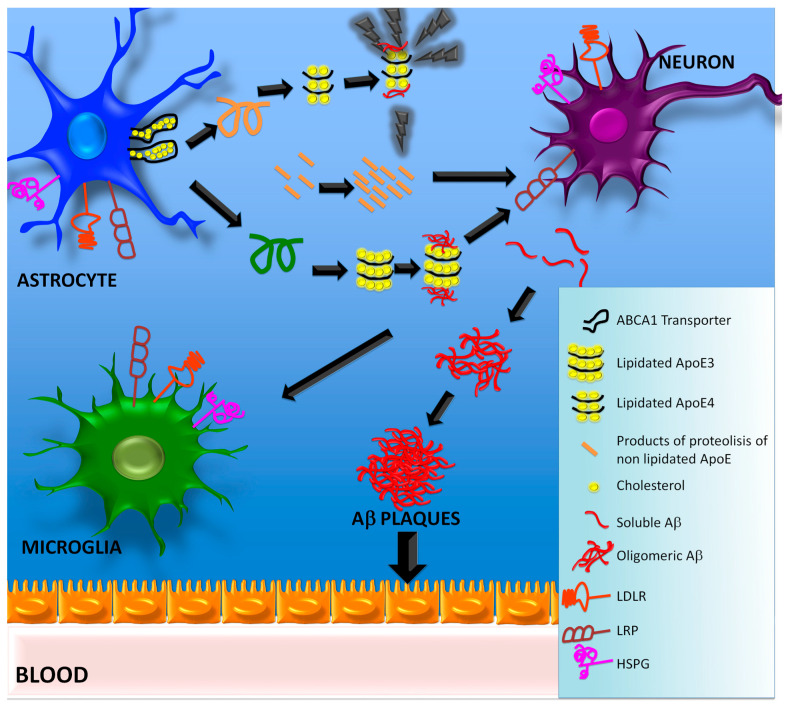
ApoE lipidation process. Astrocytic ApoE is the main source of ApoE in the brain. To accomplish its functions, after being synthesized, ApoE must be lipidated with cholesterol, triglycerides, and phospholipids. The ApoE lipidation process begins when ABCA1 translocates lipids to the plasma membrane, binds to the free form of apolipoproteins, and promotes their acceptance of the lipids translocated. The internalization of ApoE-lipid complexes by neurons is mediated by receptors like HSPG, LDLR, and LDLRP. The degree of lipidation, size, and content of ApoE-lipid complexes are isoform dependent; ApoE4 is poorly lipidated when compared to the other isoforms. When ApoE is not lipidated, it can form toxic aggregates in the neurons that can evade the secretory pathway, stimulate tau phosphorylation, and cause mitochondrial dysfunction. Lipidated ApoE has higher Aβ binding affinity than delipidated molecules, which enhances the clearance of Aβ through the blood stream and prevents its accumulation.

**Figure 3 ijms-23-12092-f003:**
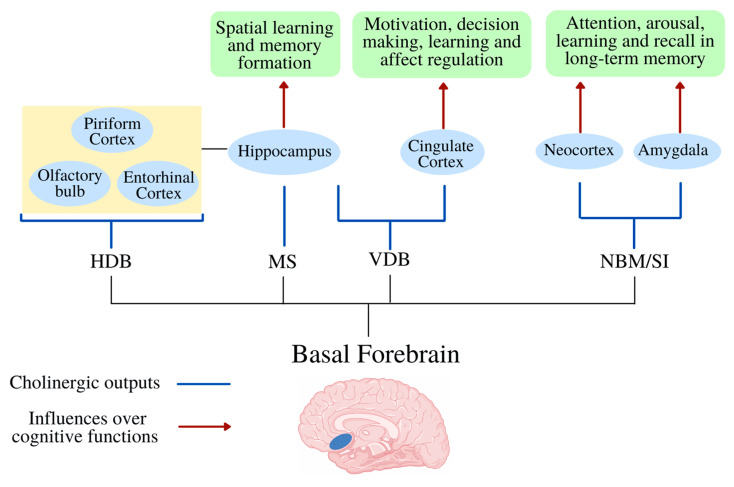
Cholinergic outputs of the Basal Forebrain. The Basal forebrain (BF) is composed of the medial septum (MS), vertical limb of the diagonal band of Broca (VDB), horizontal limb of the diagonal band of Broca (HDB), and nucleus basalis magnocellularis (NBM), or substantia innominata (SI); the Basal forebrain has many cholinergic outputs with several cortical and subcortical structures; all of which have some influence over important cognitive functions.

**Figure 4 ijms-23-12092-f004:**
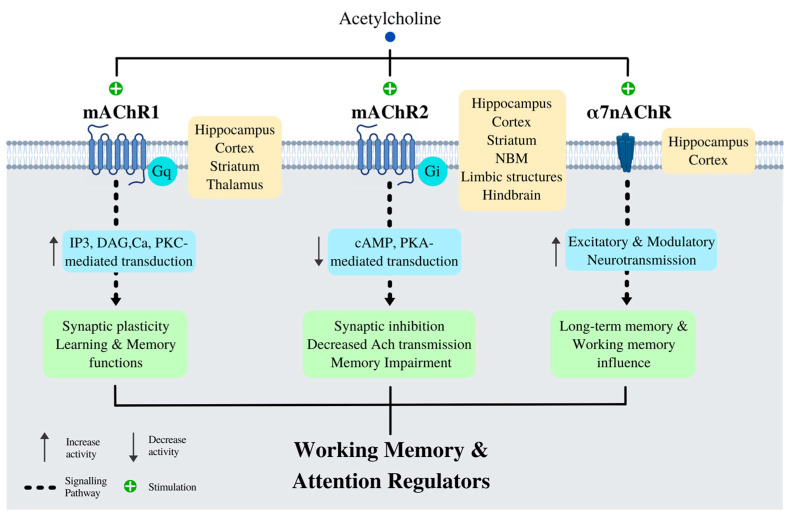
Acetylcholine receptors and their influence on Working Memory & Attention. The three main cholinergic receptors that influence working memory and attention are: The M1 muscarinic acetylcholine receptor (mAChR1), M2 muscarinic acetylcholine receptor (mAChR2), and the α7 nicotinic acetylcholine receptor (α7 nAChR). These receptors are widely distributed across cortical and subcortical structures interplaying to regulate working memory and attention.

**Figure 5 ijms-23-12092-f005:**
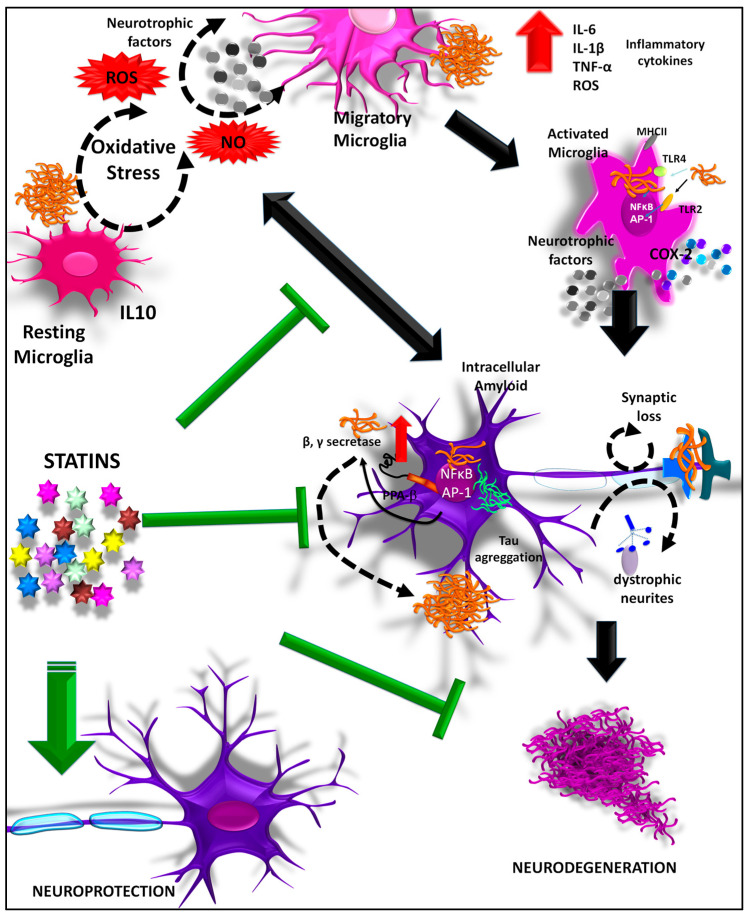
Neuroprotective effect of statins. Statins attenuate neuronal damage caused by Aβ, conducting to decreased Aβ levels and Aβ aggregation. Aβ induces tau hyperphosphorylation through activation of GSK3β, leading to neuronal death. Statin also can reduce Aβ-induced neuronal apoptosis, as well as the inflammatory processes that lead to neurodegeneration, increasing neuroprotection.

**Figure 6 ijms-23-12092-f006:**
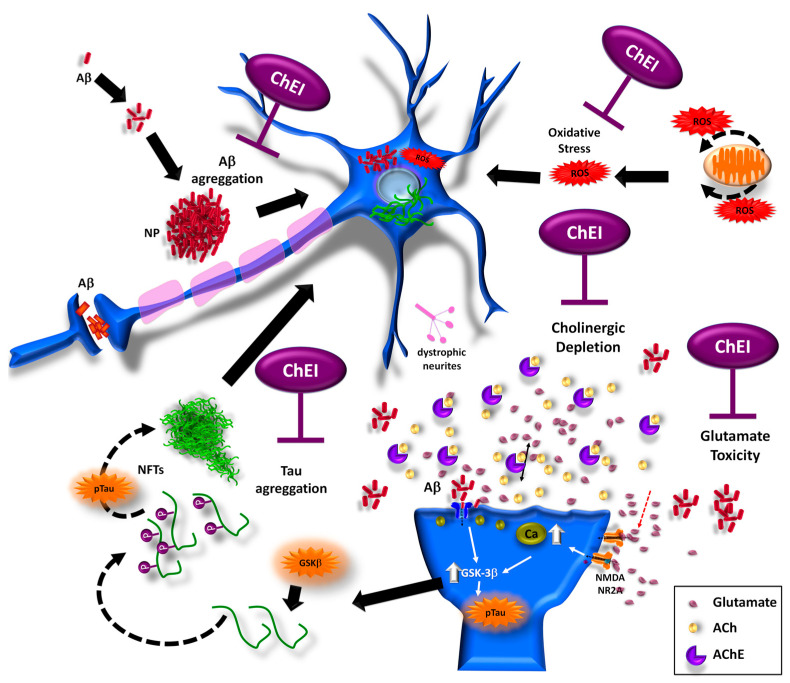
Neuroprotective effects of AChE inhibitors. Acetylcholine (ACh) is one of the main neurotransmitters, released at the synapse, and broken down by the enzyme acetylcholinesterase (AChE) into choline and acetate. The excess of AChE generates a cholinergic depletion, generating damage at the synaptic level and degeneration of cholinergic projections in the basal forebrain that leads to cognitive impairment. ChEIs reduce neuronal death caused by PNs and MNFs formation, decrease glutamate-mediated toxicity, cholinergic depletion, phosphorylated tau aggregation, ROS damage, and decrease amyloid levels.

**Figure 7 ijms-23-12092-f007:**
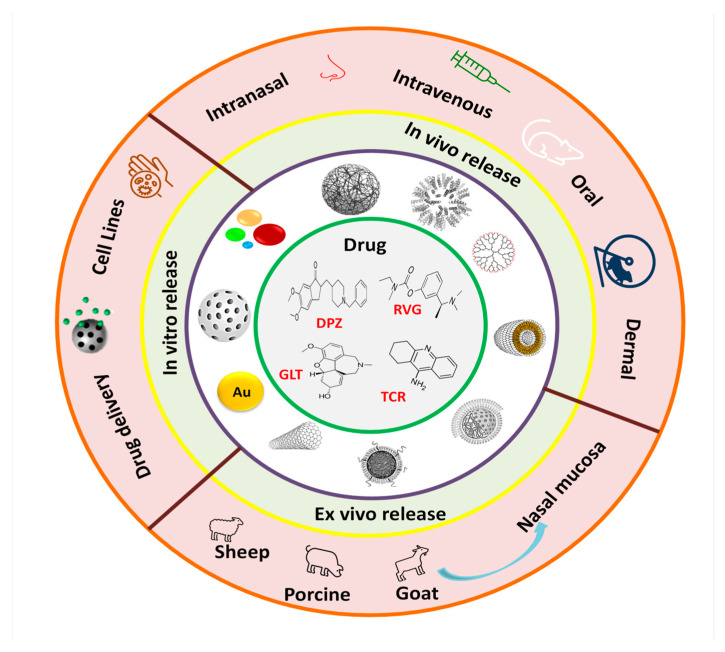
Chemical structures of the main anticholinergic drug were stabilized on the different nanomaterials for subsequent in vitro, ex vivo, and in vivo release using sheep, porcine, and goat nasal mucosa. While rats of several strains and mice were used as animal models administered by nasal, oral, transdermal, and subcutaneous methods via the drug-nanoparticles formulations. Abbreviations: DPZ (Donepezil), RVG (Rivastigmine), GLT (Galantamine), TCR (Tacrine).

**Table 1 ijms-23-12092-t001:** Potential use of statins for the treatment of AD.

Statin	Evidence	Model	Reference
**Lovastatin**	Decreases neurotoxic effect of Aβ, favors non-amyloidogenic processing, upregulates the expression of α7 nAChR and the stimulation of extracellular signal-regulated kinase phosphorylation (ERK), activates Akt signaling pathway, which in turn, inhibits downstream GSK-3β and decreases tau hyperphosphorylation and aggregation.	SH-SY5Y cells. Rat hippocampal neuronal cells	[212,214,218]
**Atorvastatin**	Improves inflammatory impairment, depresses inflammatory responses, and prevents Aβ25–35-induced neurotoxicity.	Mice injected with Aβ 25–35 intracerebroventricularly. Cultured hippocampal neurons	[217]
**Simvastatin**	Improves spatial cognitive function, protects neurogenesis through α7nAChR-cascading PI3K-Akt, increases BDNF levels, reduces neuronal apoptosis, improves functional and pathological recovery by activating the Wnt/β-catenin signaling pathway.	Mice injected with Aβ 25-35 intracerebroventricularly. Spinal cord injury rat model. APP mice model.	[213,215,216]
**Fluvastatin**	Decreases Aβ accumulation, and prevents cognitive deterioration, oxidative stress and loss of neurons in the basal forebrain induced by Aβ25-35.	Mice injected with Aβ 25-35 intracerebroventricularly.	[220,221]
**Mevastatin**	Restores insulin signaling and protects amyloid-induced neurotoxicity through activation of AMP-activated protein kinase (AMPK).	SK-N-MC neuronal cells.	[219]

## Data Availability

Not applicable.

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
