# Peer review of "Amyloid β, Lipid Metabolism, Basal Cholinergic System, and Therapeutics in Alzheimer’s Disease"

_ijms, 2022, doi:10.3390/ijms232012092_

Round 1

Reviewer 1 Report

Reviewer comments and suggestions

The review summarizes the current knowledge on the role of lipids such as cholesterol and phospholipids in Alzheimer’s disease, their relationship with the basal cholinergic system, along with potential disease-modifying therapies.

I listed below my suggestion and comments 

  1. In the abstract, Line 30-31, The authors could delete these lines as the subsequent lines meaning was similar to this sentence
  2. Line 36 The line seems to be incomplete, please explain the way or delete the "in this way"
  3. Line 56 and 70 are the authors discussing the same, so why amino acid number were differ
  4. Line 107-108 What would be the consequence of this
  5. Line 143-145 Not match with the previous paragraph, please read the previous sentence and modify
  6. Line 151-152 Could you elaborate the study and a typo error in line 153
  7. Line 158-159 need references and incomplete lines
  8. Line 172-173 Where this plasma membrane author was talking, please be specific.. I saw there was missing place/position of the membrane where authors discussed the sentence. please proofread
  9. Line 194 is this important to discussing phospholipid here
  10. Avoid long sentences such as line 296-300 and for section line 337 I suggest authors be specific to the subtitle. Discussing lots of things that were irrelevant was not appropriate. 
  11. Line 372-373 There should be a difference in the legend and the text of the manuscript
  12. Line 585 Here the authors have to make points on nanotherapy as they added sections
  13. Line 589 It would be better to present in the form of table rather than discussing in the para.. already the authors made elongated the paper 
  14. Line 591 why tacrine not discussed and line 595 make it short as the authors provided information about the other drugs
  15. Line 758 The title is short, please modify it and other subtitle should be merged for a better look of the manuscript, just my suggestions. No need to exaggerate the section, more reader-friendly we have to make the manuscript.
  16. Please check the MDPI journal style for using the journal name. I show there was a mistake in the format, please modify it

Author Response

We appreciate the comments made by the reviewer. The requested corrections are in red.
 A table was included outlining the effects of different statins, as suggested by the reviewer.

Reviewer 2 Report

The review highlights the influence of lipids on Alzheimer's disease and their relationship with the basal cholinergic system. The authors summarize the noncovalent interactions between the lipids and proteins, which is essential for the understanding of the pathology and helpful for finding a new therapeutic strategy for Alzheimer's regime. The authors also have concluded some nanotherapeutic strategies. Considering the significance and systematicness of the current review, I support the publication of this work in the International Journal of Molecular Sciences, if the following comment has been considered.

For the diagnosis and treatment of Alzheimer's disease, the supramolecular strategies based on molecular recognition are increasingly popular (e.g., Nature Chem. 2019 11, 86–93. and Adv. Mater., 2021, 33, 2006483.) the corresponding investigation had better been included in the review.

Author Response

Thank you very much for the comments made by the reviewer. According to his suggestions, the English was revised again and the requested information was added.

Reviewer 3 Report

The manuscript by Campos-Peña et al. describes the current knowledge about the involvement of amyloid b peptide, lipid metabolism, and cholinergic system in Alzheimer's disease. Furthermore, currently used and experimental drugs against this disease are also covered. The review is comprehensive and very well written. It can constitute a valuable article for readers of the International Journal of Molecular Sciences.

There is only one issue that needs to be corrected. In lines 56 and 77, the authors wrote, respectively: "NPs are constituted of a 40-42 amino acid peptide, known as amyloid-b Ab," and "The most common Aβ found aggregated in neuritic plaques of brain Alzheimer’s patients, is the Aβ42." However, a growing new literature shows that shorter, N-truncated Ab peptides are more abundant than Ab40 and Ab42. It relates to both healthy and AD individuals. It has to be updated in the manuscript appropriately.

Author Response

Thank you very much for the comments made by the reviewer. Regarding what it mentions about the N-truncated Ab isoforms, a brief description was added. The purpose of our work lies in the Ab 1-42 peptide, which is why we had excluded information